# Research on the impact path of virtual streamers characteristics on agricultural product consumers' purchase intention

Lin Jiang◯\*, Min Li

School of International Economics and Trade, Fujian Business University, Fuzhou, FuJian, China

\* 379024429@qq.com

## Abstract

Live-streaming e-commerce for agricultural product has become a new exploration in rural revitalization and digital commerce in China, with AI-driven virtual streamers emerging as a new trend in this field. Focusing on the context of virtual streamers marketing, this study constructs a relationship model between virtual streamers characteristics and consumers' purchase intention for agricultural products, using human-machine trust as a moderating variable and communication presence and emotional presence as mediating variables. The results show that the affinity, anthropomorphism, professionalism, and responsiveness of virtual streamers can positively influence consumers' purchase intention for agricultural products through the mediating effects of communication presence and emotional presence.Both communication presence and emotional presence positively affect consumers' purchase intention. Additionally, human-machine trust plays a significant moderating role in the relationship between virtual streamers characteristics and social presence. This study provides insights for the development of virtual streamers in the agricultural industry and offers practical and effective pathways to enhance the quality and efficiency of live-streaming e-commerce for agricultural products.

## Introduction

The No.1 Central Document of 2025 proposes to develop distinctive rural industries, thoroughly implement integrated rural and agricultural development projects, foster new rural industries and businesses, and promote the high-quality development of rural e-commerce.Against the backdrop of the deepening integration of rural revitalization and digital poverty alleviation in China, live-streaming e-commerce has emerged as an important channel for agricultural product sales.In 2023, the national rural online retail sales reached 2.49 trillion yuan, a year-on-year increase of 12.5%. In 2022, the number of rural online merchants (online stores) nation-wide reached 17.303 million, of which 5.732 million were engaged in live-streaming

**Data availability statement:** All relevant data are within the manuscript and its Supporting Information files.

**Funding:** The National Social Science Foundation General Project of China (24BJY160) Fujian Natural Science Foundation General Project(2023J011143) The funders had no role in study design, data collection and analysis, decisionto publish, or preparation of the manuscript.

**Competing interests:** No authors have competing interests

e-commerce,accounting for 33.1%.Live-streaming e-commerce for agricultural products not only facilitates the sales of agricultural products and promotes farmer entrepreneurship and employment but also serves as a crucial engine for creating a new model of "digital commerce for rural revitalization" [1]. Therefore,how to scientifically utilize live streaming to support agriculture, continuously enhance consumers' purchase intentions, and thereby boost the sales of agricultural products, has become an important theoretical and practical issue in the field of rural revitalization and the digital economy.

The rapid development of artificial intelligence technology has enabled virtual digital humans not only more realistic in appearance and movement but also significantly improved in terms of interaction capabilities and intelligence levels.This technological breakthrough has made virtual streamers the fastest-growing segment in the live streaming industry.At the same time,competition among live streamers promoting agricultural products is intensifying. Some farmers show low willingness to participate in live streaming, and top-tier agricultural product streamers are becoming a scarce resource. An increasing number of agricultural enterprises are beginning to consider introducing virtual streamers as replacements for human streamers in live broadcasts. Virtual streamers offer unique advantages such as 24 h a day uninterrupted service, precise recommendation algorithms, high exclusivity and adaptability, and no risk of "scandal", providing a novel shopping experience for agricultural product consumers [2–3]. However, as a cutting-edge product of artificial intelligence technology,virtual streamers still have significant shortcomings in terms of anthropomorphism,expression richness,interactivity, and intelligence level [4–6]. Most virtual streamers for agricultural products can only mechanically read product selling points during live streams, with monotonous explanations and uniform scripts, lacking emotional and attitudinal transmission, and weak user interaction and product experience.These limitations somewhat hinder the establishment of strong emotional connections and trust relationships between virtual streamers and agricultural product consumers, leading to low purchase intentions among consumers.Therefore,while maintaining the efficiency advantages of virtual streamers, how to better improve the live-streaming effects of virtual streamers and enhance the purchase intentions of agricultural product consumers has become an issue that agricultural-related enterprises must consider when adopting virtual streamers.

From an academic research perspective, studies on virtual streamers are relatively scarce, primarily focusing on technical aspects such as key technology standards [7–8], or qualitatively analyzing the development history, current status, application strategies, and prospects of virtual streamers [9–10]. There is limited use of quantitative methods to analyze virtual streamers.By analyzing existing quantitative research on virtual streamers,it has been found that factors such as the cuteness, authenticity, responsiveness, intelligence, sociability, positive emotions, sales-driven emotions, and reaction capabilities of virtual streamers have a positive impact on consumer behavior [2,11,12]. However, research on the individual characteristics of virtual streamers by scholars is relatively fragmented, with few studies systematically organizing the key characteristics of virtual streamers, and even fewer exploring how

these characteristics influence the purchase intentions of agricultural product consumers from the perspective of consumer social interaction perception. Therefore, this study will use quantitative research methods to delve into the impact of virtual streamers characteristics on consumer purchase intentions and their mechanisms from the perspective of social interaction perception.This study makes significant theoretical contributions by establishing a systematic framework for characterizing virtual streamers, elucidating the key psychological mechanisms underlying consumer interactions with different virtual streamers characteristics, and expanding the theoretical boundaries of virtual streamers research, Social Presence Theory, and human-machine trust studies.This study focuses on addressing the following three key issues, First, how does the characteristic of virtual streamers affect the purchase intentions of agricultural product consumers, and through what aspects and pathways does it affect? Second, how do consumers perceive social interaction with virtual streamers of different characteristics, and does this affect their purchase intentions for agricultural products? Third, is the impact of virtual streamers characteristics on the purchase intentions of agricultural product consumers moderated by other factors?

## Review of related research

### Characteristics of virtual streamers

With the significant improvement in the realism, interactivity, and real-time feedback capabilities of virtual digital human technology, virtual digital humans have become a new trend in the era of the digital economy. Virtual streamers, also known as digital human anchors, refer to virtualized personas driven by technologies such as artificial intelligence. These personas are designed based on real streamers, incorporating features such as voice, appearance, language style, facial expressions, gestures, and interaction methods [13]. Virtual streamers not only possess a human-like appearance but can also recognize external environments and interact with humans, significantly evoking emotional responses and behavioral patterns in people.

Existing research has shown that the characteristics of real streamers have a direct impact on consumers' sharing and purchasing intentions, and virtual streamers similarly influence consumers' emotional and behavioral reactions [2]. Therefore, this study focuses on analyzing the impact path of virtual streamers characteristics on consumers' purchase intentions.Through a review of the literature, research on virtual streamers characteristics has evolved from holistic to granular perspectives, progressing from functional attributes to emotional dimensions, demonstrating continuous theoretical deepening.Early research primarily focused on macro-level interactive characteristics, emphasizing how virtual streamers' interactivity and social attributes enhanced consumers' perceived value and purchase intention [14]. Subsequent research shifted toward examining individual functional characteristics, identifying that anthropomorphism, responsiveness, professionalism, and intelligence could improve purchase intention through mediators factors like social presence, perceived usefulness, and immersive experience [7,15]. Recent investigations have delved deeper into emotional characteristics, revealing the distinct impact of cuteness, affinity,dynamism, and entertainment value on consumers' emotional responses [11–13].

From the perspective of stylistic classification, virtual endorsers can be divided into humanoid designs and anime-like designs, which demonstrate distinct levels of affinity. Advertising research has established a significant positive correlation between virtual endorsers' affinity and their commercial communication efficacy [16]. Given the functional similarity between virtual streamers and virtual endorsers, we adopt affinity as a key metric for evaluating virtual streamers characteristics. Commerce research has demonstrated that human streamers' professionalism significantly enhances consumers' purchase intention [17]. Extending this finding to virtual streaming contexts, this research will likewise examine the impact of virtual streamers' professionalism on consumers' purchase intention. Existing research consistently confirms that virtual streamers' anthropomorphism and responsiveness capabilities exert significant positive effects on consumers' purchase intention [12,13,18].

Meanwhile, Fiske's warmth-capability dual-dimension framework of social cognition posits that when evaluating individuals or groups, people primarily rely on two fundamental dimensions: warmth and capability. At the same time, Gao et al. and Gong Xiaoxiao both believe that the impact of virtual streamers on consumer behavioral intentions can be analyzed from the dimensions of warmth and capability [13,19]. Combining the above empirical results, we will focus on the impact of virtual streamers characteristics on consumers' purchasing intentions from the dimensions of warmth and capability. The warmth dimension refers to the emotional warmth that virtual streamers bring to consumers during communication, which can be divided into affinity and anthropomorphism indicators; the capability dimension refers to a series of capabilities or skills that virtual streamers possess during the live broadcast process, which can be divided into professionalism and responsiveness indicators.Affinity measures the degree to which a virtual streamer is perceived as approachable, friendly and pleasant. Anthropomorphism measures the extent to which a virtual streamer resembles real humans in appearance and vocal style. Professionalism measures the depth of the virtual streamers 'ability to analyze specialized domains and products.Responsiveness measures the virtual streamers' capability to quickly respond to consumer needs and provide timely services.

## Social Presence

Social presence refers to the degree to which a person is perceived as a "real person" and the salience of interpersonal relationships with others when using media for communication [20]. From the perspective of consumers' perception of social interaction, social presence focuses on the role of media and explores the extent to which consumers establish interpersonal relationships such as warmth, familiarity, intimacy, and pleasure with others through media, reflecting an emotional of co-presence with others. Social presence is an important construct in mediated environments and has gradually become a research hotspot in fields such as online shopping and online learning, closely linked to outcomes such as driving trust, purchase intention, and continuous usage intention [21]. Currently, few scholars have focused on the social presence of consumers interacting with virtual streamers, and there is a lack of integrated analysis of the impact of virtual streamers characteristics on consumer purchase intention from the perspective of social presence. Therefore, it is necessary to study the connections between virtual streamers characteristics, social presence, and consumer purchase intention. Meanwhile, with the continuous change of scenarios, social presence has also evolved from a single-dimensional division to two-dimensional, three-dimensional, and even multi-dimensional detailed divisions [20–21]. In the context of virtual streamers, consumers can engage in real-time, smooth communication with virtual streamers through methods such as bullet comments and live connections. At the same time, the virtual streamer's appearance, language style, professional competence, real-time changes in viewer numbers, likes, sending virtual gifts, and affirmative comment content can all create pleasurable experiences for consumers, promoting emotional presence.Therefore,the division of social presence into communication and emotional dimensions can be adopted for analysis.

## Research hypotheses and model construction

### Research hypotheses

**The impact of virtual streamers characteristics.** In the warmth dimension,affinity refers to the sense of closeness, friendliness, and pleasantness conveyed by virtual streamers during their interactions with consumers, embodying a human-like warmth [4,22]. Affinity is a force that draws people closer, making them willing to engage, and it forms consumers' positive first impressions of virtual streamers. From a psychological perspective, people tend to trust things that appear approachable and friendly. The stronger the affinity of a virtual streamer, the more it can eliminate consumers' sense of unfamiliarity towards the streamer, making consumers more willing to communicate and interact with the streamer, thereby establishing good interpersonal relationships and increasing trust [23–24]. Therefore, we propose the following hypothesis:

H1a: Affinity is positively correlated with communication presence.
H1b: Affinity is positively correlated with emotional presence.

In the warmth dimension,anthropomorphism refers to the degree to which a virtual streamer resembles a real human in terms of appearance and speech style, emphasizing the lifelikeness of the virtual streamers [18]. When a virtual streamer is endowed with human-like appearance characteristics, consumers are more likely to perceive them as a real, emotional individual, thereby attracting consumer attention and favor, and promoting purchase intention [13,25]. In addition to visual appearance, when a virtual streamer can communicate with consumers in a natural and fluent manner using anthropomorphic tone, pitch, and speech rate, it further solidifies the streamer's personal identity and sense of social belonging, thereby evoking stronger feelings of familiarity and trust in consumers [26]. A highly anthropomorphic virtual streamer can eliminate the sense of distance between consumers and others in the live stream, making consumers willing to communicate and establish close emotional connections with the streamer, thus generating creates a sense of warmth and authenticity in interpersonal interactions, thereby enhancing purchase intention. Therefore, we propose the following hypotheses:

H2a: Anthropomorphism is positively correlated with communication presence.

H2b: Anthropomorphism is positively correlated with emotional presence.

In the capability dimension, professionalism refers to the virtual streamers' professional competence, which stems from their mastery of the professional field and knowledge of the live-streamed products [12]. Research has found that the professionalism of virtual streamers can lead consumers to perceive varying degrees of identification or compliance at a psychological level, thereby directly influencing their emotions, cognition, and behavior [13,27]. Influenced by factors such as information cocoons and information asymmetry, most consumers lack comprehensiveness, objectivity, and innovation in their decision-making. If virtual streamers can leverage artificial intelligence technology to obtain and display detailed product information from multiple angles and various channels, consumers will psychologically identify with the streamer and actively engage in communication,thereby significantly reducing uncertainty during the shopping process and enhancing purchase intention. Therefore, we propose the following hypothesis:

H3a: Professionalism is positively correlated with communicative presence.

H3b: Professionalism is positively correlated with emotional presence.

In the capability dimension,responsiveness refers to the virtual streamers' ability to quickly respond to consumer needs and provide timely services. Responsiveness generates interactive feedback for consumers, arousing their desire to participate, altering their cognition and emotions, and enhancing their sense of participation and control [12,28]. Benefiting from the responsiveness of virtual streamers, they can promptly reply to and interact with consumers, creating a sense of human contact and a real-world shopping experience, thereby shortening the psychological distance between them. Research has found that responsiveness can enhance consumers' perceived value and experience, reduce perceived risk and psychological distance [29–30]. As responsiveness continues to strengthen, consumers increasingly feel the warmth of the virtual streamers, becoming more willing to actively communicate with them, and are more likely to form positive emotions and attitudes, thereby generating purchase intention. Therefore, we propose the following hypotheses:

H4a: Responsiveness is positively correlated with communication presence.

H4b: Responsiveness is positively correlated with emotional presence.

**The impact of social presence.** With the advancement of artificial intelligence technology, virtual streamers are becoming increasingly similar to real individuals in terms of appearance, voice style, and interaction. The deeper the consumers' perception of the virtual streamers' real existence, the stronger their sense of social presence becomes. Communication presence refers to the convenience and fluidity with which consumers feel they can communicate and interact with others in the virtual streamers' live broadcast room [20]. When individuals with professional expertise or high credibility engage in smooth and timely communication with consumers through product explanations, bullet comments,and live interactions, it creates a sense of real shopping for consumers.This helps foster positive emotional experiences and a pleasant purchasing sensation, thereby influencing consumers' purchase intentions. Emotional presence refers to the degree of emotional interaction between consumers and others in the virtual streamers' live broadcast room [21]. Virtual streamers can evoke a sense of social existence in consumers through lifelike appearances,

strong affinity, friendly and warm communication, and a professional and reliable image.This makes consumers feel accepted by a certain organization, group, or relationship, thereby shortening the psychological distance between them and consumers.Additionally, environmental stimuli in the live broadcast room, such as real-time changes in viewer numbers, likes, virtual gifts, and positive comments, can enhance consumers' sense of pleasure. Supported by pleasant emotions, emotional presence is generated, which in turn facilitates the formation of purchase intentions. Therefore, we propose the following hypotheses:

H5a: Communication presence is positively correlated with consumers' purchase intentions.

H5b: Emotional presence is positively correlated with consumers' purchase intentions.

H6a: Communication presence mediates the effect of virtual streamers characteristics on consumers' purchase intentions.

H6b: Emotional presence mediates the effect of virtual streamers characteristics on consumers' purchase intentions.

**The moderating role of human-machine trust.** Human-machine trust refers to the degree of trust users have in artificial intelligence systems or machines. With the proliferation of AI systems and machines, there exists a gap in people's understanding of the practical application capabilities of intelligent technologies. Trust becomes a crucial means to bridge this gap. In dynamic and unstructured environments, the level of trust users have in AI systems or machines significantly impacts the quality of human-machine interaction and collaboration [31]. The process of establishing human-machine trust essentially reflects an interactive process between people's psychological expectations of intelligent technologies and the optimization efficacy of the technologies themselves [32]. Consumers with high levels of human-machine trust perceive AI systems or machines as unbiased and reliable, capable of exhibiting human-like characteristics such as providing valuable responses and adhering to the reciprocity mechanisms in social interactions.Conversely,consumers with low human-machine trust believe that artificial intelligence systems or machines still differ from the important roles played by humans.They doubt that AI systems or machines can help users solve complex problems or meet personalized needs. In the context of e-commerce virtual streamers, consumers with high levels of human-machine trust are more likely to trust virtual streamers, engage in interactive communication with them, and immerse themselves in the scenario of interacting with virtual streamers.In this case, characteristics of virtual streamers such as professionalism and responsiveness can provide consumers with a stronger sense of presence, thereby fostering purchase intentions.On the other hand, consumers with low levels of human-machine trust, regardless of the anthropomorphism, interactivity, or professionalism of the virtual streamers, tend to have a stronger sense of self-control.They often lack the motivation and energy to deeply understand the virtual streamers, making it difficult to establish deep interactions and emotional resonance with them. Therefore, the following hypotheses are proposed:However, consumers with low human-machine trust, regardless of the anthropomorphism, interactivity, or professionalism of virtual streamers, tend to have a stronger sense of self-control. They often lack the motivation and energy to deeply understand virtual streamers and find it difficult to engage in deep interaction and emotional resonance with them. Therefore, the following hypotheses are proposed:

H7a: human-machine trust positively moderates the impact of virtual streamers characteristics on communication presence.

H7b: human-machine trust positively moderates the impact of virtual streamers characteristics on emotional presence.

## Model construction

In this study, human-machine trust is introduced as a moderating variable, while the affinity, anthropomorphism, professionalism, and responsiveness of virtual streamers serve as independent variables. The sense of social presence, specifically communication presence and emotional presence, acts as mediating variables, and the purchase intention of agricultural product consumers is the dependent variable. Based on this framework, a model is constructed to examine the impact of virtual streamers characteristics on the purchase intention of agricultural product consumers. The specific model is shown in Fig 1.

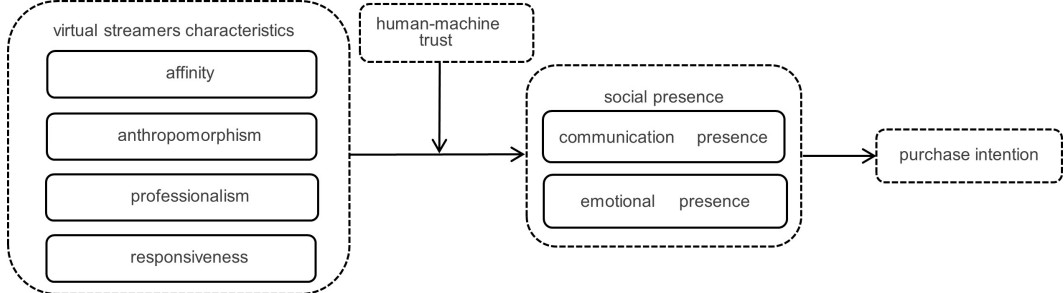

**Fig 1. Model of the impact of virtual streamers characteristics on agricultural product consumers' purchase intention.**

## Research design

### Scale design

All measurement items in the scale were adapted from mature scales used in existing research, while also being modified and refined to fit the context of agricultural product live-streaming e-commerce. To ensure the validity of the questionnaire, after drafting the initial version, we first consulted two rural e-commerce experts to revise the questionnaire items. Based on their feedback, we invited 30 users who had watched virtual streamers live streams to participate in a pre-test. Combining reliability and validity analysis with user feedback on the logic, language, and other aspects of the questionnaire, we further revised and refined the items, ultimately forming the final questionnaire.The online survey consisted of three parts:The online questionnaire contained three sections. In the first section, we introduced the purpose of the research and promised to keep the respondents' information safe and anonymous. In addition, we described virtual streamers in live streaming commerce to ensure the respondents comprehended our research context. In the second section, to guarantee the validity of the questionnaire, we set a pre-screening question to ask the consumers whether they had watched a virtual streamers in live streaming commerce. Only those who answered "yes" were permitted to participate in the survey. In the second section, we collected the respondents' demographic information, including their gender, age, education, and income.The third part is the main section, the qualiffed respondents answered the questions about the research constructs based on their latest viewing experience of virtual streamers in a live streaming commerce.All scales used a 5-point Likert scale, where 5 indicated "strongly agree," 3 indicated "neutral," and 1 indicated "strongly disagree."

### Data sources and descriptive statistics

The study relied solely on human expertise and research techniques to conduct the study, without utilizing any AI-based methods to complete the paper. Ethical review and approval was not required for the study on human participants in accordance with the local legislation.This study did not involve human participants, animal experiments, or sensitive data requiring ethical approval.

The questionnaire was distributed and data was collected through the Chinese internet research platform,Wenjuanxing (https://www.wjx.cn/),utilizing) platforms such as WeChat, QQ, Xiaohongshu, and Weibo. Due to the relatively small number of consumers who watch virtual live streams of agricultural products, a snowball sampling procedure was used to collect data. To better identify potential valid respondents, after respondents who had watched virtual live streams of agricultural products completed the survey, they were invited to share the questionnaire link with other consumers who had similar experiences.A total of 340 questionnaires were distributed from May 1 to August 30, 2024, and 38 invalid questionnaires were excluded due to reasons such as duplicate responses, incomplete answers, contradictory responses, not having watched virtual live streams of agricultural products, and excessively short response times.Ultimately, 302 valid questionnaires were obtained, resulting in an overall effective response rate

of 88.82%.We did not have access to any information that could identify individual participants. All data were anonymized before analysis.Descriptive statistical analysis was conducted on the basic information in the valid questionnaires, as shown in Table 1. Overall, the sample data's gender and age structure are similar to the internet user attribute structure distribution reported in the "55th Statistical Report on Internet Development in China" released by the China Internet Network Information Center in January 2025.The sample structure is reasonable and aligns with reality, making it suitable for subsequent data processing.

## Empirical results and analysis

### Reliability and validity tests

To ensure the rationality and validity of the empirical research, AMOS 24.0 was used to test the reliability and validity of the model variables. Cronbach's α coefficient and Construct Reliability (CR) were used to test the internal consistency reliability of the scale. The results of the reliability test are shown in Table 2. The standard factor loadings of all indicator variables are distributed between 0.70 and 0.90. The Cronbach's α coefficients of all latent variables in the scale are greater than 0.79, proving that all indicator variables have high scientific validity.The convergent validity (AVE) and composite reliability (CR) of each dimension were higher than the minimum thresholds of 0.5 and 0.7, respectively, indicating good convergent validity and composite reliability.

In the confirmatory factor analysis (CFA) test, the results of discriminant validity are shown in Table 3. There are significant correlations between the dimensions (p < 0.01), and the correlation coefficients are all smaller than the corresponding square roots of the AVE, indicating good discriminant validity among the variables of each dimension.

### Variable correlation analysis

Using SPSS 26, the Pearson correlation coefficients between the characteristics of virtual streamers, communication presence, emotional presence, and the purchase intention of agricultural product consumers were calculated, as shown in Table 4. The characteristics of virtual e-commerce streamers are significantly positively correlated with the dependent variable, the purchase intention of agricultural product consumers, and are significant at the 1% level. The mediating variables, communication presence and emotional presence, are positively correlated with the purchase intention of agricultural product consumers at the 1% confidence level. This preliminarily validates the rationality of the proposed hypothesis variables and sets the stage for subsequent hypothesis testing.

Table 1. Descriptive analysis of demographic variables.

| Measurement Indicator | Indicator Classification | Frequency | Proportion (%) | Measurement Indicator | Indicator Classification | Frequency | Proportion (%) |
|---|---|---|---|---|---|---|---|
| **Gender** | Male | 134 | 44.37 | **Personal Monthly Disposable Income** | Under 3000 yuan | 40 | 13.25 |
| | Female | 168 | 55.63 | | 3001–5000 yuan | 92 | 30.46 |
| **Age** | Under 20 years old | 29 | 9.60 | | 5001–8000 yuan | 110 | 36.42 |
| | 21–30 years old | 93 | 30.79 | | 8001–10000 yuan | 31 | 10.26 |
| | 31–40 years old | 108 | 35.76 | | 10001 yuan and above | 29 | 9.61 |
| | 41–50 years old | 35 | 11.59 | **Educational Level** | Under High school | 18 | 5.96 |
| | 51–60years old | 21 | 6.95 | | Junior College | 89 | 29.47 |
| | 61 years old and above | 16 | 5.31 | | Undergraduate | 187 | 61.93 |
| Effective Sample Size: 302 | | | | | Postgraduate and above | 8 | 2.64 |

## Goodness-of-fit test

AMOS 24.0 was used to test the overall model, employing the maximum likelihood estimation method to substitute the corresponding data into the research model. The analysis results of the fit indices are shown in Table 5. All indices fall within a reasonable range, indicating that the model has a good fit.

## Structural equation model test

Path analysis was conducted based on the research hypotheses, resulting in Fig 2 and Table 6. The results show that affinity, professionalism, anthropomorphism, and responsiveness have significant positive effects on communication presence, with corresponding p-values all less than 0.05 and standardized coefficients of 0.205, 0.248, 0.278, and

**Table 2. Convergent validity and composite reliability tests for each dimension.**

| Variable | Measurement Item | Factor Loading | AVE | CR | Cronbach's a | Reference Source |
|---|---|---|---|---|---|---|
| **Affinity (AF)** | AF1 | 0.703 | 0.578 | 0.804 | 0.803 | VERHAGEN(2014) [22] |
| | AF2 | 0.795 | | | | |
| | AF3 | 0.779 | | | | |
| **Anthropomorphism (AN)** | AN1 | 0.774 | 0.566 | 0.796 | 0.793 | Zhan Ying(2024) [18] |
| | AN2 | 0.782 | | | | |
| | AN3 | 0.797 | | | | |
| **Professionalism (PR)** | PR1 | 0.772 | 0.593 | 0.814 | 0.813 | Li Rong(2025) [12] |
| | PR2 | 0.788 | | | | |
| | PR3 | 0.750 | | | | |
| **Responsiveness (RE)** | RE1 | 0.855 | 0.654 | 0.849 | 0.848 | Gao Tingting (2024) [31] |
| | RE2 | 0.843 | | | | |
| | RE3 | 0.721 | | | | |
| **Communication presence (CP)** | CP1 | 0.804 | 0.583 | 0.807 | 0.808 | Zhao Hongxia et al.(2015) [20] |
| | CP2 | 0.748 | | | | |
| | CP3 | 0.737 | | | | |
| **Emotional presence (EP)** | EP1 | 0.751 | 0.679 | 0.863 | 0.861 | Yu Xin (2021) [21] |
| | EP2 | 0.866 | | | | |
| | EP3 | 0.850 | | | | |
| **Purchase intentions (PI)** | PI1 | 0.783 | 0.650 | 0.847 | 0.844 | Ou et al.(2014) [33] |
| | PI2 | 0.845 | | | | |
| | PI3 | 0.788 | | | | |

**Table 3. Discriminant validity analysis.**

| | AF | AN | PR | RE | CP | EP | PI |
|---|---|---|---|---|---|---|---|
| **AF** | **0.578** | | | | | | |
| **AN** | 0.349 | **0.593** | | | | | |
| **PR** | 0.180 | 0.312 | **0.566** | | | | |
| **RE** | 0.127 | 0.235 | 0.243 | **0.654** | | | |
| **CP** | 0.386 | 0.497 | 0.444 | 0.422 | **0.583** | | |
| **EP** | 0.310 | 0.424 | 0.389 | 0.307 | 0.462 | **0.679** | |
| **PI** | 0.408 | 0.545 | 0.501 | 0.403 | 0.568 | 0.565 | **0.650** |

**Table 4. Correlation analysis of variables.**

|  | AF | AN | PR | RE | CP | EP | PI |
|---|---|---|---|---|---|---|---|
| AF | 1 |  |  |  |  |  |  |
| AN | .291** | 1 |  |  |  |  |  |
| PR | .146* | .254** | 1 |  |  |  |  |
| RE | .096* | .181** | .199** | 1 |  |  |  |
| CP | .311** | .396** | .357** | .322** | 1 |  |  |
| EP | .247** | .371** | .326** | .262** | .393** | 1 |  |
| PI | .339** | .458** | .418** | .338** | .492** | .473** | 1 |

Note: *p<0.05, **p<0.01, ***p<0.001.

**Table 5. Fit of the measurement model.**

| Fit Index | Optimal Fit Range | Computed Value | Fit Status |
|---|---|---|---|
| CMIN/DF | <3 | 1.422 | Good |
| NFI | >0.9 | 0.919 | Good |
| GFI | >0.9 | 0.932 | Good |
| AGFI | >0.9 | 0.908 | Good |
| RMSEA | <0.08 | 0.037 | Good |
| IFI | >0.9 | 0.975 | Good |
| CFI | >0.9 | 0.974 | Good |

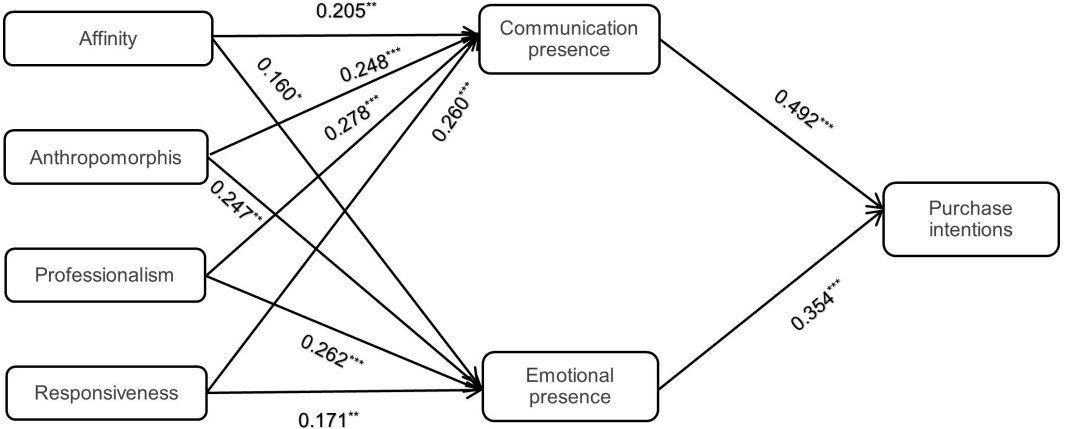

**Fig 2. Structural Equation Model Diagram.** Note: *p<0.05, **p<0.01, ***p<0.001.

0.260.Therefore, hypotheses H1a, H2a, H3a, and H4a are supported. Affinity,professionalism, anthropomorphism, and responsiveness also have significant positive effects on emotional presence, with corresponding p-values all less than 0.05 and standardized coefficients of 0.160, 0.247, 0.262, and 0.171.Thus, hypotheses H1b, H2b, H3b, and H4b are supported. Communication presence and emotional presence have significant positive effects on purchase intention, with corresponding p-values less than 0.05 and standardized coefficients of 0.492 and 0.354.Therefore, hypotheses H5a and H5b are supported.

**Table 6. Results of hypothesis testing (N = 302).**

| Hypothesis | Mediation Path | β | S.E. | C.R. | P | Remark |
|---|---|---|---|---|---|---|
| H1a | Affinity→Communication presence | 0.205 | 0.075 | 3.179 | ** | Supported |
| H1b | Affinity→Emotional presence | 0.160 | 0.085 | 2.398 | * | Supported |
| H2a | Anthropomorphism→Communication presence | 0.248 | 0.090 | 3.642 | *** | Supported |
| H2b | Anthropomorphism→Emotional presence | 0.247 | 0.100 | 3.587 | ** | Supported |
| H3a | Professionalism→Communication presence | 0.278 | 0.068 | 3.902 | *** | Supported |
| H3b | Professionalism→Emotional presence | 0.262 | 0.076 | 3.625 | *** | Supported |
| H4a | Responsiveness→Communication presence | 0.260 | 0.065 | 4.206 | *** | Supported |
| H4b | Responsiveness→Emotional presence | 0.171 | 0.072 | 2.718 | ** | Supported |
| H5a | Communication presence→Purchase intentions | 0.492 | 0.096 | 6.907 | *** | Supported |
| H5b | Emotional presence→Purchase intentions | 0.354 | 0.082 | 5.320 | *** | Supported |

Note: *p < 0.05, **p < 0.01, ***p < 0.001.

## Mediation effect test of social presence

The mediation effect was tested using the BootStrapping method in AMOS.A total of 5,000 bootstrap samples were randomly drawn from the original data using repeated random sampling. The estimates were made using the 2.5th and 97.5th percentiles, with a 95% confidence interval. The mediation effects of communication presence and emotional presence are shown in Table 7. The 95% confidence intervals of the mediation effects do not include 0, indicating significant mediation effects. Therefore, hypotheses H6a and H6b are supported.

## Moderation Effect Test of Human-Machine Trust

When testing the moderation effect of human-machine trust, the collinearity relationships among the variables need to be considered. The independent variables (the four characteristics of virtual streamers),the moderating variable (human-machine trust), and the mediating variables (communication presence and emotional presence) were mean-centered. A hierarchical regression approach was used, with the centered independent variables and moderating variable set as the first layer, and the interaction terms of the centered independent variables and moderating variable set as the second layer. The results of the moderation effect test of human-machine trust are shown in Table 8. After including the interaction terms, all regression coefficients are significant at the 0.01 level, and the $R^2$ values increase, indicating enhanced

**Table 7. Bootstrap analysis of mediating effects.**

| Mediation Path | SE | 95% BC-CI | Results |
|---|---|---|---|
| Affinity→Communication presence→Purchase intentions | 0.037 (0.022) | [0.002,0.087] | Significant |
| Anthropomorphism→Communication presence→Purchase intentions | 0.044 (0.025) | [0.004,0.101] | Significant |
| Professionalism→Communication presence→Purchase intentions | 0.049 (0.028) | [0.004,0.110] | Significant |
| Responsiveness→Communication presence→Purchase intentions | 0.048 (0.026) | [0.005,0.105] | Significant |
| Affinity→Emotional presence→Purchase intentions | 0.035 (0.021) | [0.001,0.084] | Significant |
| Anthropomorphism→Emotional presence→Purchase intentions | 0.053 (0.023) | [0.013,0.104] | Significant |
| Professionalism→Emotional presence→Purchase intentions | 0.057 (0.028) | [0.012,0.120] | Significant |
| Responsiveness→Emotional presence→Purchase intentions | 0.037 (0.021) | [0.002,0.085] | Significant |

Note: BC-CI = bias-corrected confidence interval.

**Table 8. Moderating effect analysis of human-machine trust.**

| Term | Communication presence | | Emotional presence | |
|---|---|---|---|---|
| | Model 1: Adding Moderator | Model 2: Adding Interaction | Model 3: Adding Moderator | Model 4: Adding Interaction |
| **Affinity** | 0.571** | 0.554** | 0.547** | 0.550** |
| **Anthropomorphism** | 0.422** | 0.429** | 0.171** | 0.179** |
| **Professionalism** | 0.500** | 0.484** | 0.422** | 0.428** |
| **Responsiveness** | 0.316** | 0.319** | 0.370** | 0.379** |
| **Affinity*Human-machine trust** | | 0.233*** | | 0.193** |
| **Affinity*Human-machine trust ($R^2$)** | 0.417 | 0.451 | 0.470 | 0.487 |
| **Anthropomorphism* Human-machine trust** | | 0.201** | | 0.297* |
| **Anthropomorphism* Human-machine trust ($R^2$)** | 0.351 | 0.398 | 0.339 | 0.412 |
| **Professionalism* Human-machine trust** | | 0.094* | | 0.087** |
| **Professionalism* Human-machine trust ($R^2$)** | 0.473 | 0.489 | 0.343 | 0.417 |
| **Responsiveness* Human-machine trust** | | 0.172*** | | 0.197** |
| **Responsiveness* Human-machine trust ($R^2$)** | 0.311 | 0.330 | 0.370 | 0.379 |

Note: *$p < 0.05$, **$p < 0.01$, ***$p < 0.001$.

explanatory power and better model fit. This suggests that human-machine trust has a positive moderating effect between the characteristics of virtual streamers and social presence. Therefore, hypotheses H7a and H7b are supported.

## Conclusions and recommendations

### Research conclusions

This study focuses on the new model of digital commerce aiding rural revitalization through live streaming, specifically examining the impact of virtual streamers characteristics on the purchase intentions of agricultural product consumers. Based on actual questionnaire data, with human-machine trust as the moderating variable and communication presence and emotional presence as mediating variables, the study reveals the mechanisms and effects of virtual streamers characteristics on the purchase intentions of agricultural product consumers. The conclusions are as follows:

Virtual streamers characteristics influence mediating variables:The characteristics of virtual streamers, including affinity, anthropomorphism, professionalism and responsiveness, can affect the mediating variables of communication presence and emotional presence among agricultural product consumers, thereby stimulating their purchase intentions. Among these, professionalism has the greatest impact on communication presence, followed by responsiveness, anthropomorphism, and affinity. Similarly, professionalism has the greatest impact on emotional presence, followed by anthropomorphism, responsiveness, and affinity.Specifically, when virtual streamers can leverage artificial intelligence technology to provide objective, professional, and detailed product information from multiple perspectives and through various channels, consumers will psychologically perceive a sense of identification or compliance. They will also actively communicate with the virtual streamers, thereby aiding their purchase decision-making and enhancing their purchase intention.The anthropomorphism of virtual streamers is reflected in the vivid realism of their appearance, vocal style, movements, and other characteristics. The more pronounced these features are, the more they can

create an authentic and immersive viewing experience for consumers, who will be more willing to engage in communication and establish a close emotional connection with the virtual streamers.The more promptly virtual streamers address consumer needs, the more intelligent they are perceived. When both their communication and emotional needs are met, consumers' positive attitudes toward the virtual streamers and their purchase intention for the products will also increase accordingly.The stronger the affinity of virtual streamers, the more pronounced the sense of closeness, friendliness, and authenticity they create. The more consumers watch virtual streamers' live broadcast, the more they will experience feelings of pleasure and trust, making them more willing to purchase the products recommended by the virtual streamers.

Mediating variables positively influence purchase intentions: Both communication presence and emotional presence positively influence the purchase intentions of agricultural product consumers, with communication presence having a greater impact than emotional presence.On one hand, in e-commerce live streams, virtual streamers can effectively meet consumers' information needs by presenting products through explanations, real-time chat interactions, and live linking, thereby reducing uncertainty in their decision-making process and enhancing purchase intention. This aligns with the core premise of Social Presence Theory—when people perceive the existence of another "social entity", they generate positive emotional responses. On the other hand, as live streams interactions deepen, the emotional connection between consumers and virtual streamers strengthens, fostering greater enjoyment and harmony. Over time, this cultivates an intimate and warm social relationship within the live streams environment. The higher the level of consumers' emotional presence, the more their purchasing decisions are influenced by this atmosphere.

Human-machine trust plays a moderating role: Human-machine trust significantly moderates the relationship between virtual streamers characteristics and the mediating variables of communication presence and emotional presence. The higher the level of the consumer's human-machine trust, the more pronounced the positive impact of virtual streamers characteristics on purchase intentions.For consumers with higher human-machine trust, they are more receptive to the virtual streamers' affinity, anthropomorphism, professionalism, and responsiveness, making it easier for them to establish trust in the virtual streamers. For consumers with lower human-machine trust, more time and information may be needed to build trust in virtual streamers as an emerging technological product.

## Theoretical contributions

The findings of this study carry significant implications in multiple aspects.First, by extending the research scope of e-commerce streamers from human streamers to virtual streamers, this study substantially enriches the literature on live streaming commerce. Previous research in this domain has predominantly focused on how characteristics of human streamers influence consumer behavior, with few studies systematically exploring the influence of key individual characteristics of virtual streamers on consumer purchase intention.This study examines the impact of virtual streamers characteristics on consumers' purchasing intentions from the dimensions of warmth and capability, thereby broadening the scope of research on virtual streamers. Second, adopting the Social Presence Theory perspective to explore the mechanism through which virtual streamers characteristics influence consumer's purchase intention, this study robustly validates the applicability of Social Presence Theory in explaining purchase behavior in AI-driven live streaming contexts. By investigating the mediating roles of communication presence and emotional presence between virtual streamers characteristics and consumer's purchase intention, this study provides theoretical guidance for cultivating social presence in virtual streamer interactions. Furthermore, this study extends the application boundaries of human-machine trust theory in commercial settings while offering a novel theoretical perspective for understanding consumer decision-making in human-AI interaction scenarios. The research reveals that human-machine trust plays a crucial moderating role in the relationship between virtual streamers characteristics and social presence, thereby providing richer theoretical dimensions for explaining consumer' online behavior in human-AI interaction contexts.

## Practical contributions

By examining the impact of virtual streamer characteristics on consumers' purchase intentions for agricultural products, this study provides insights for the development of virtual streamers in the agricultural sector.

Enhance affinity and anthropomorphism of virtual streamers.Affinity: Agricultural enterprises can utilize artificial intelligence (AI) voice technology, virtual reality, and motion capture to optimize features such as clothing, hairstyles, and facial expressions, emphasizing a friendly and lively personality to continuously improve the affinity of virtual streamers. By enhancing the sense of closeness, friendliness, and joy during interactions, consumers can experience a stronger sense of social presence.In terms of image design, enterprises should fully consider consumers' aesthetic preferences and psychological needs.Using advanced AI technologies to meticulously design the facial features and expressions of virtual streamers to make them more lifelike and reduce the psychological distance for consumers.Language Style: advanced speech synthesis and affective computing technologies should be utilized to make language expression more natural, fluent, and emotional, thereby enhancing the realism and humanization of virtual streamers.

Improve professionalism and real-time responsiveness of virtual streamers.On one hand, by optimizing artificial intelligence algorithms, enterprises can enhance the virtual streamer's expertise in agricultural product knowledge,shopping guidance,and language charm.Virtual streamers should provide in-depth and detailed introductions to the features and advantages of agricultural products in live streams, improving the ability to manage the live stream and handle unexpected situations. This will create high-quality, professional virtual streamers that are trusted by consumers and welcomed by platforms.On the other hand, through natural language processing and emotion recognition technologies, virtual streamers should be able to accurately understand consumer intentions and emotions, responding more quickly and directly to consumer inquiries.Additionally, Enterprises should also focus on the design and use of interactive tools such as live stream comments to promptly address consumer inquiries, enhancing the convenience of user interactions and strengthening users' sense of communication presence and emotional presence.

Implement differentiated and personalized marketing strategies.Agricultural enterprises should pay close attention to consumers' levels of human-machine trust and implement differentiated marketing strategies for users with varying levels of trust.For consumers with high levels of human-machine trust, enterprises should emphasize the unique appeal of virtual streamers in agricultural product live streaming, leveraging their advantages and innovative features to provide personalized and intelligent service experiences.For consumers with low levels of human-machine trust,enterprises should invite them to participate in virtual streamers live streams through user-friendly interfaces, blind box draws, and coupon distributions to gradually increase their trust.Simultaneously,enterprises should create personalized shopping scenarios and design engaging and entertaining live stream topics to attract consumer attention. Use creative methods to present products, continuously meeting consumers' personalized needs.

## Limitations and future research directions

This study has several limitations. First, we employed a survey method to collect data on consumers' actual perceptions of virtual streamers. Future studies should further expand the sample size and enhance its diversity of the samples. Additionally, experimental methods such as eye-tracking experiments could be employed to ensure the broader applicability and accuracy of the research findings. Second, numerous factors influence consumers' purchase intentions in the context of virtual streamers livestream marketing. This study only focused on the impact of the virtual streamers' individual characteristics. Future studies could explore the effects of product and scenario factors, as well as their alignment with virtual streamers on purchase intentions. Additionally, the study of virtual streamers characteristics could be refined from perspectives such as the virtual streamers' intelligence, information quality, and popularity.Third, in the mediation analysis, only the mediating effect of social presence on the relationship between virtual streamers characteristics and consumers' purchase intentions was examined. Future studies could explore other mediating variables, such as flow experience, consumer emotions, consumer trust, and perceived risk.In the moderation effect test, this study solely investigated the

influence of human-machine trust. Future studies should comprehensively consider various potential moderating variables, such as consumers' education level, age, gender, and occupation, to enhance the precision and reliability of the research.Fourth, although virtual streamers livestream marketing is widely adopted in Chinese e-commerce and shows a global expansion trend, this study focused exclusively on the Chinese cultural context.Future research could examine our conceptual model and hypotheses in other cultural contexts.

## Supporting information

**S1 Data.**
(XLS)

## Author contributions

**Conceptualization:** Min Li.

**Data curation:** Lin Jiang.

**Formal analysis:** Lin Jiang.

**Funding acquisition:** Lin Jiang.

**Investigation:** Lin Jiang.

**Methodology:** Lin Jiang.

**Project administration:** Lin Jiang.

**Resources:** Lin Jiang.

**Software:** Lin Jiang.

**Supervision:** Lin Jiang.

**Validation:** Lin Jiang.

**Visualization:** Lin Jiang.

**Writing – original draft:** Lin Jiang.

**Writing – review & editing:** Min Li.

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
