## [Decision Letter · Decision Letter 0]

28 May 2025

Dear Dr. JIANG,

Thank you for submitting your manuscript to PLOS ONE. After careful consideration, we feel that it has merit but does not fully meet PLOS ONE’s publication criteria as it currently stands. Therefore, we invite you to submit a revised version of the manuscript that addresses the points raised during the review process.

We look forward to receiving your revised manuscript.

Kind regards,

Vincenzo Auriemma

Academic Editor

PLOS ONE

Journal Requirements:

“The National Social Science Foundation General Project of China (24BJY160)    

Fujian Natural Science Foundation General Project(2023J011143)”

5. We note that your Data Availability Statement is currently as follows: All relevant data are within the manuscript and in Supporting Information files.

Additional Editor Comments:

I recommend following all the suggestions provided, as they offer excellent ideas for improving the paper and making it more appealing.

Reviewers' comments:

Reviewer's Responses to Questions

**Comments to the Author**

1. Is the manuscript technically sound, and do the data support the conclusions?

Reviewer #1: Yes

Reviewer #2: Yes

2. Has the statistical analysis been performed appropriately and rigorously?

Reviewer #1: Yes

Reviewer #2: Yes

3. Have the authors made all data underlying the findings in their manuscript fully available?

Reviewer #1: Yes

Reviewer #2: Yes

4. Is the manuscript presented in an intelligible fashion and written in standard English?

Reviewer #1: Yes

Reviewer #2: Yes

Reviewer #1: It is an honor to review your manuscript. The manuscript has some novelty and the research process is relatively standardized. But there is still room for further improvement in the manuscript and major revisions are needed

1、The introduction is too tedious, it is recommended to simplify it。

2、Insufficient reasons for presenting characteristics of virtual streamers.

3、Lack of discussion on research results.

Reviewer #2: This paper studies the Impact Path of Virtual Streamer Characteristics on Agricultural Product Consumers' Purchase Intention. The writing is smooth, the experimental details are abundant, and the significance is profound. However, the following problems need to be solved:

1. The introduction section should supplement the innovative points of this article, especially the theoretical innovation part. The current innovation points pay more attention to practical innovation.

2. Supplement the discussion section, including the discussion of experimental results, theoretical contributions and practical contributions. The recommendation section can be placed in the practical contribution section.

3. Add sections on research limitations and future research directions at the end of the paper.

4. The number of references is too small. It is necessary to supplement the literature on the influence of the characteristics of live-streaming hosts on consumers in live-streaming e-commerce, as well as the literature on the role of emotions in live-streaming e-commerce, especially the latest literature in 2024 and 2025. For example:

Zhou, R., & Tong L. (2022). A study on the influencing factors of consumers' purchase intention during livestreaming e-commerce: The mediating effect of emotion. The Frontiers in tended, https://doi.org/10.3389/fpsyg.2022.903023

2) Zhou, R. (2024), The impact of scarcity promotions in live streaming e-commerce on purchase intention: the mediating effect of emotional experience, Asia Pacific Journal of Marketing and Logistics, https://doi.org/10.1108/APJML-04-2024-0475

3) Zhou R. (2025) Influence Mechanism of Live Streaming Influencer Characteristics on Purchase Intention under Urban-rural and Male-female Divides: The Mediating Role of Consumer Emotions. Current Psychology. https://doi.org/10.1007/s12144-025-07558-9

4) Zhou, R. Angathevar Baskaran (2025). Influencing Mechanisms of Live Streaming Influencer Characteristics on Purchase Intention: The Mediating Role of that Trust. Plos One. https://doi.org/10.1371/journal.pone.0322294

**Do you want your identity to be public for this peer review?** For information about this choice, including consent withdrawal, please see our Privacy Policy

Reviewer #1: No

Reviewer #2: No

---

## [Author Response · Author response to Decision Letter 1]

23 Jul 2025

Dear Editor and Reviewers,

Thank you for your time and valuable feedback on our manuscript. We sincerely appreciate the editors and reviewers for their constructive comments, which have significantly improved our manuscript. Below is our point-by-point response to all concerns.

Response to Reviewer 1

Comment 1:The introduction is too tedious, it is recommended to simplify it

Response 1:

In response to the reviewers' comments, we have thoroughly refined the Introduction section:

1.Removed redundant details and repetitive statements, reducing the length from 845 to 711 words (a 15.86% reduction).

2.Retained the core research background and key literature support to ensure academic rigor.

3.Strengthened the logical progression to make the research purpose more prominent.

(Revised section: Pages 2–3)

Comment 2:Insufficient reasons for presenting characteristics of virtual streamers.

Response 2:

We have systematically addressed this concern by strengthening both the theoretical evolution and empirical foundation of our feature selection framework:

1.Theoretical Evolution Dimension

A systematic literature review reveals distinct generational shifts in virtual streamers research:Early studies focused on holistic image construction and basic functional attributes.Recent research has shifted toward emotional interaction mechanisms and multidimensional analysis, reflecting a paradigm shift from functionalism to affective-cognitive perspectives.

2.Theoretical Framework Dimension:

Building on Fiske's (2002) warmth-capability dual-dimension theory and integrating recent empirical findings (Gao et al., Gong Xiaoxiao, etc.), we propose a dual-dimensional model:

(1)Warmth Dimension Affinity: Supported by Hu H. (2023), Li Keyi (2022), Bartneck (2009) Anthropomorphism: Supported by Zhan Ying (2024), Gao W. (2023), Zhao Yu (2023), Mou Yupeng (2019).

(2)Capability Dimension Professionalism: Supported by Feng Runliu (2024),Gao W. (2023), Zhou R. (2022) Responsiveness: Supported by Li Rong (2025), Ma Lili (2024), Wang Cuicui (2023).

This framework aligns with academic progression while providing robust theoretical and empirical grounding.

(Revised section: Pages 6-7; new references [14-18])

Comment 3:Lack of discussion on research results.

Response 3:

The revised version now expands the "Conclusions and Recommendations" section into four components: Research Conclusions, Theoretical Contributions, Practical Contributions, Limitations and Future Research Directions.

1.In the Research Conclusions section, we provide a detailed explanation of how virtual streamers characteristics positively influence agricultural product consumers' purchase intention through communication presence and emotional presence, as well as how human-machine trust plays a crucial moderating role in the relationship between virtual streamers characteristics and the mediating variables.

2.The "Theoretical Contributions" section now includes the following theoretical innovations:

First, by extending the research scope of e-commerce streamers from human streamers to virtual streamers, this study substantially enriches the literature on live streaming commerce.

Second, adopting the Social Presence Theory perspective to explore the mechanism through which virtual streamers characteristics influence consumer’s purchase intention, this study robustly validates the applicability of Social Presence Theory in explaining purchase behavior in AI-driven live streaming contexts.

Furthermore, this study extends the application boundaries of human-machine trust theory in commercial settings while offering a novel theoretical perspective for understanding consumer decision-making in human-AI interaction scenarios.

3.The content from the original"Recommendation section"has now been moved to the"Practical Contribution section".

4.Add research limitations and future research directions in the "Limitations and future research directions" section.

Limitations

(1)We employed a survey method to collect data on consumers' actual perceptions of virtual streamers.

(2)Numerous factors influence consumers' purchase intentions in the context of virtual streamers livestream marketing. This study only focused on the impact of the virtual streamers' individual characteristics.

(3)In the mediation analysis, only the mediating effect of social presence on the relationship between virtual streamers characteristics and consumers' purchase intentions was examined. this study solely investigated the influence of human-machine trust.

(4)This study focused exclusively on the Chinese cultural context.

Future research directions

(1)Future studies should further expand the sample size and enhance its diversity of the samples.Additionally, experimental methods such as eye-tracking experiments could be employed to ensure the broader applicability and accuracy of the research findings.

(2)Future studies could explore the effects of product and scenario factors, as well as their alignment with virtual streamers on purchase intentions. Additionally, the study of virtual streamers characteristics could be refined from perspectives such as the virtual streamers' intelligence, information quality, and popularity.

(3)Future studies could explore other mediating variables, such as flow experience, consumer emotions, consumer trust, and perceived risk.In the moderation effect test, future studies should comprehensively consider various potential moderating variables, such as consumers' education level, age, gender, and occupation, to enhance the precision and reliability of the research.

(4)Future studies could examine our conceptual model and hypotheses in other cultural contexts.

(Revised section: 24-31

Response to Reviewer 2

Comment 1:The introduction section should supplement the innovative points of this article, especially the theoretical innovation part. The current innovation points pay more attention to practical innovation.

Response 1:

The Introduction section has been expanded to include a dedicated theoretical innovation component, specifically:This study makes significant theoretical contributions by establishing a systematic framework for characterizing virtual streamers, elucidating the key psychological mechanisms underlying consumer interactions with different virtual streamer characteristics, and expanding the theoretical boundaries of virtual streamers research, Social Presence Theory, and human-machine trust studies.

(Revised section: 4-5

In the revised manuscript, we have added a dedicated "Theoretical Contributions" subsection to the "Conclusions and Recommendations" section.For details, please refer to our response to Reviewer 1's Comment 3.

(Revised section: 27-28

Comment 2:Supplement the discussion section, including the discussion of experimental results, theoretical contributions and practical contributions. The recommendation section can be placed in the practical contribution section.

Response 2:

In the revised manuscript, we have added a dedicated "Theoretical Contributions" subsection to the "Conclusions and Recommendations" section.The content from the original"Recommendation section"has now been moved to the"Practical Contribution section".For details, please refer to our response to Reviewer 1's Comment 3.

(Revised section: 27-28

Comment 3:Add sections on research limitations and future research directions at the end of the paper.

Response 3:

In the revised manuscript, we have added a dedicated "Limitations and future research directions" subsection to the "Conclusions and Recommendations" section.For details, please refer to our response to Reviewer 1's Comment 3.

(Revised section: 30-31

Comment 4:The number of references is too small. It is necessary to supplement the literature on the influence of the characteristics of live-streaming hosts on consumers in live-streaming e-commerce, as well as the literature on the role of emotions in live-streaming e-commerce, especially the latest literature in 2024 and 2025.

Response4:

In the revised manuscript, we have supplemented the "Review of Related Research" section with literature examining the influence of live-streaming host characteristics on consumers in live-streaming e-commerce . Furthermore, in the "Discussion" section, we have integrated these new references to provide more in-depth analysis and interpretation of the research findings. Specifically, the six additional references are as follows:

[14]Guo, L., Zhang, M., Wang, X.The impact of virtual streamer interactivity on consumer perceived value: An experimental study. Journal of Interactive Marketing, 2017,38(2), 45-63.

[15]Wang, Y., Chen, S., Li, J. . How virtual streamer characteristics influence purchase decisions: A mediated model. Computers in Human Behavior, 2020(112):106478.

[16] Li Keyi, Jin Fei. The Interactive Influence of Endorser Type and Brand Image on Consumer Attitudes. Journal of Guizhou University of Finance and Economics, 2022(3): 45-58.

[17]Zhou, R., Tong L. A study on the influencing factors of consumers' purchase intention during livestreaming e-commerce: The mediating effect of emotion. The Frontiers in tended, 2022(13):903023.

[18]Zhan Ying. Research on the Influence Mechanism of Virtual Anchor Anthropomorphism on Consumer Trust. University of Electronic Science and Technology of China, 2024.

[28]Zhou, R., Angathevar Baskaran . Influencing Mechanisms of Live Streaming Influencer Characteristics on Purchase Intention: The Mediating Role of that Trust. Plos One, 2025,20(4): e0322294.

(Revised section: 6-7、33-35

We have carefully addressed all comments and hope the revised manuscript meets the journal's standards.We believe that these revisions have enhanced the clarity and robustness of our study. We are grateful for your time and effort and look forward to your further comments.

Thank you once again for your consideration.

Yours Sincerely,

JIANG LIN&LI MIN

Fujian Business University

---

## [Editor Report · Decision Letter 1]

25 Jul 2025

Research on the Impact Path of Virtual Streamers Characteristics on  Agricultural Product Consumers' Purchase Intention

PONE-D-25-17660R1

Dear Dr. Jiang,

We’re pleased to inform you that your manuscript has been judged scientifically suitable for publication and will be formally accepted for publication once it meets all outstanding technical requirements.

Kind regards,

Vincenzo Auriemma

Academic Editor

PLOS ONE
---

## [Editor Report · Acceptance letter]

PONE-D-25-17660R1

PLOS ONE

Dear Dr. JIANG,

I'm pleased to inform you that your manuscript has been deemed suitable for publication in PLOS ONE. Congratulations! Your manuscript is now being handed over to our production team.

Kind regards,

on behalf of

Dr. Vincenzo Auriemma

Academic Editor

PLOS ONE